# Learning Altruistic Behaviours in Reinforcement Learning without External Rewards

**Tim Franzmeyer**
University of Oxford
frtim@robots.ox.ac.uk

**Mateusz Malinowski**
DeepMind
mateuszm@google.com

**João F. Henriques**
University of Oxford
joao@robots.ox.ac.uk

## Abstract

Can artificial agents learn to assist others in achieving their goals without knowing what those goals are? Generic reinforcement learning agents could be trained to behave altruistically towards others by rewarding them for altruistic behaviour, i.e., rewarding them for benefiting other agents in a given situation. Such an approach assumes that other agents' goals are known so that the altruistic agent can cooperate in achieving those goals. However, explicit knowledge of other agents' goals is often difficult to acquire. In the case of human agents, their goals and preferences may be difficult to express fully; they might be ambiguous or even contradictory. Thus, it is beneficial to develop agents that do not depend on external supervision and learn altruistic behaviour in a task-agnostic manner. We propose to act altruistically towards other agents by giving them more choice and allowing them to achieve their goals better. Some concrete examples include opening a door for others or safeguarding them to pursue their objectives without interference. We formalize this concept and propose an altruistic agent that learns to increase the choices another agent has by preferring to maximize the number of states that the other agent can reach in its future. We evaluate our approach in three different multi-agent environments where another agent's success depends on altruistic behaviour. Finally, we show that our unsupervised agents can perform comparably to agents explicitly trained to work cooperatively, in some cases even outperforming them.

## 1 Introduction

Altruistic behaviour is often described as behaviour that is intended to benefit others, sometimes at a cost for the actor (Dowding and Monroe, 1997; Fehr and Fischbacher, 2003). Such behaviour is a desirable trait when integrating artificial intelligence into various aspects of human life and society – such as personal artificial assistants, house or warehouse robots, autonomous vehicles, and even recommender systems for news and entertainment. By observing and interacting with us, we may expect that artificial agents could adapt to our behaviour and objectives, and learn to act helpfully and selflessly. Altruistic behaviour could be a step towards value alignment (Allen et al., 2005; Gabriel, 2020), which aims to incorporate common-sense human values into artificial agents.

Typically, we could achieve such an altruistic behaviour through various forms of supervision such as providing ground-truth actions at each time step, training agents with reinforcement learning (RL) and suitable rewards, or through imitation learning (Song et al., 2018). However, none of the approaches above scale up easily. They either require a large amount of supervision or carefully crafted rewards that can easily be misstated, leading to unwanted behaviour (Russell, 2019, ch. 1).

How can one agent support another agent without knowing its goals? One clue might be the instrumental convergence hypothesis (Bostrom, 2017; Omohundro, 2008; Russell, 2019), which states that intelligent agents with varied goals are likely to pursue common subgoals which are generally useful (instrumental). Some examples are resource acquisition, cognitive enhancement or self-preservation, which all increase an agent's chance of achieving almost arbitrary final goals. This hypothesis has been validated theoretically under many models, including resource games (Benson-Tilsen and Soares, 2016) and large classes of policies in discrete MDPs (Turner et al., 2019).

While instrumental convergence is central to the discussion of value alignment and safe AI (Bostrom, 2017), since many instrumental subgoals have harmful effects, we believe that it is also a key to supporting agents with ill-defined goals and values, such as humans. The reason is that enabling instrumental subgoals for other agents (or not impeding them) can be beneficial, for a wide variety of goals and preferences. Since these subgoals occur frequently for rational agents, enabling them has the highest chance of success in the absence of more information about the other agent's preferences, even if it is not guaranteed in the worst case.

We speculate that having the ability to reach many future states is one of the most general convergent subgoals. It subsumes self-preservation (avoiding absorbent states), resource acquisition (if they are prerequisites to some actions), and generally maintaining the ability to pursue many goals. There is theoretical evidence that many optimal agents pursue this subgoal (Turner et al., 2019) (see sec. 3.2). Thus, we propose to train agents to support other agents by maximizing their *choice* (future state availability). This unsupervised approach learns altruistic behaviour without any extrinsic supervision such as rewards or expert demonstrations.

We evaluate our methods in three diverse multi-agent environments. We always assume there are at least two agents: the leader agent that executes its own policy and can be trained using standard supervised methods, and an altruistic agent whose role is to help the leader. The performance of the altruistic agent is thus defined as the reward (success) achieved by the leader agent. In all our environments, the overall success of the leader agent depends on the altruistic agents' behaviour. We show that our unsupervised approach outperforms unsupervised baselines by a large margin and, in some cases, also outperforms the supervised ones. Finally, we demonstrate possible failure cases of our approach where maximising the leader agent's choice can lead to suboptimal behaviour.

Our work makes the following three contributions:

- We devise a multi-agent RL framework for intrinsically motivated artificial agents that act altruistically by maximising the choice of others.
- We define and evaluate three task-agnostic methods to estimate the choice that an agent has in a given situation, which are all related to the variety in states it can reach.
- We experimentally evaluate our unsupervised approach in three multi-agent environments and are able to match and, in some cases, outperform supervised baselines.

## 2 RELATED WORK

To the best of our knowledge, we are the first to experimentally evaluate unsupervised agents with purely altruistic objectives. However, there are many related concepts in the literature.

In *human-robot cooperation*, a robotic agent aids a human agent in achieving its goals (Pérez-D'Arpino and Shah, 2015; Hadfield-Menell et al., 2016; Baker et al., 2006; Dragan and Srinivasa, 2013; Fisac et al., 2017; 2020; Javdani et al., 2015; Dragan and Srinivasa, 2013; Macindoe et al., 2012; Pellegrinelli et al., 2016). Methods from *Inverse RL* (IRL) are often employed to infer human goals, which are then utilized by the robot agent to support the human. IRL itself aims to learn objectives from observations and can be used in single-agent (Fu et al., 2017) and multi-agent scenarios (Song et al., 2018; Yu et al., 2019; Jeon et al., 2020). However, IRL relies on the existence of expert demonstrations, which are often difficult to get at scale. In complex environments, it also often suffers from ambiguity of solutions (Arora and Doshi, 2021).

In single-agent reinforcement learning, *empowerment* – which measures an agent's capacity to affect its environment (Klyubin et al., 2005; 2008) – is used to enable intrinsically-motivated exploration (Gregor et al., 2016; Volpi and Polani, 2020). Empowerment is also used for multi-agent cooperation (Guckelsberger et al., 2016; Du et al., 2020). Du et al. (2020) use empowerment to develop a helper agent that assists a (simulated) human agent by maximizing the human's empowerment, constituting the research work most similar to ours. In contrast to our approach, it requires privileged access to an environment simulator and therefore does not allow to learn helpful or altruistic behaviour only from observation. Furthermore, the approach is not unsupervised.

There are also mathematical formalizations of *instrumental convergence* (Bostrom, 2017). Benson-Tilsen and Soares (2016) analyze a MDP that makes finite resource allocation explicit, and find that optimal agents with arbitrary reward functions tend to deplete available resources. Turner

et al. (2019) propose "power" as a convergent subgoal, which they define as the average difference between the state value of an optimal policy and the reward in the same state. They show that, for environments with certain symmetries, a larger proportion of optimal agents prefer states with higher power. In sec. 3.2 we will describe these symmetries and relate the result to our method.

## 3 METHODS

In this section, we formalize our framework. We start with the generic definition describing multi-agent setting. Next, we describe our framework where we show various approaches to estimate choice for a single agent, and how it can be applied to a two-agents Markov Game.

**Markov Game.** We consider a Markov Game (Littman, 1994), which generalizes a Markov Decision Process (MDP) to a multi-agent scenario. In a Markov Game, agents interact in the same environment. At time step $t$, each agent (the $i$th of a total of $N$ agents) takes the action $a_i^t$, receives a reward $r_i^t$, and finally the environment transitions from state $s^t$ to $s^{t+1}$. A Markov Game is then defined by a state space $\mathcal{S}$ ($s^t \in \mathcal{S}$), a distribution of initial states $\eta$, the action space $\mathcal{A}_i$ ($a_i^t \in \mathcal{A}_i$) and reward function $r_i(s, a_1, \ldots, a_N)$ of each agent $i$, an environment state transition probability $P(s^{t+1}|s^t, a_1, \ldots, a_N)$, and finally the agents' discount factors $\gamma_i$.

### 3.1 ESTIMATING CHOICE FOR A SINGLE AGENT

We first consider a single-agent scenario, i.e. $N = 1$, where only a leader agent, indicated by the subscript $L$, interacts with the environment through its pretrained stochastic policy $\pi_L$. We assume that the leader acts Boltzmann-rationally, i.e. that it chooses high-value actions with higher probability. We believe this to be a reasonable assumption, as, in comparison to deterministic policies, stochastic policies are more robust (Zhang et al., 2020), and often achieve better results in real-world-alike partially observable stochastic domains (Kaelbling et al., 1998).

We denote the leader agent's generic choice in a given state $s$ as $C_L(s)$, for which we propose concrete realizations below. Each method relies on the random variable $S^{t+n}$, with values $s^{t+n} \in \mathcal{S}$, which refers to the leader agent's state after $n$ environment transitions from a starting state $s_t$. Its probability mass function is defined as the $n$-step state distribution of the underlying single-agent MDP, conditioned on the current state: $p(s^{t+n}|s^t) = P(S^{t+n} = s|\pi_L, s^t)$.

**Discrete choice.** Our first derived method simply defines the choice of the leader agent in state $s^t$ as the number of states that it can reach within $n$ transitions, which we refer to as its *discrete choice*:

$$DC_L^n(s^t) = |\text{range}\left(S^{t+n}|s^t\right)|, \tag{1}$$

where $\text{range}(X)$ is the set of all values that a random variable $X$ takes on with positive probability and $|\cdot|$ measures the size of that set. While this count-based estimator of choice is intuitive and easily interpretable, it can hardly be estimated practically in large or continuous state spaces. It also discards information about the probability of reaching these states.

**Entropic choice.** It can be shown that the entropy of a random variable $X$ acts as a lower bound for the size of the set of values that $X$ takes on with positive probability (Galvin, 2014, Property 2.6), i.e. $H(X) \leq \log |\text{range}(X)|$. We define a lower bound of the *discrete choice* by computing the Shannon entropy of the $n$-step state distribution, which we refer to as the agent's *entropic choice*:

$$EC_L^n(s^t) = H(S^{t+n}|s^t) = -\sum_{s \in S} P(S^{t+n} = s|\pi_L, s^t) \log\left(P(S^{t+n} = s|\pi_L, s^t)\right), \tag{2}$$

which estimates the agent's choice as the variety in its state after $n$ transitions. Unlike eq. 1, $EC_L^n$ can be computed in continuous state spaces or efficiently estimated by Monte Carlo sampling.

**Immediate choice.** To further simplify entropic choice and reduce its computational complexity, we may limit the look-ahead horizon to $n = 1$ and assume an injective relationship from actions to states, i.e. no two actions taken at $s^t$ lead to the equivalent state $s^{t+1}$. This assumption is often true in navigation environments, where different step-actions result in different states. We can then simplify the one-step state distribution of the leader agent to $p(s^{t+n}|s^t) = P(S^{t+1} = s|\pi_L, s^t) = \pi(a_L^t = a|s^t)$, and compute a simplified, short-horizon entropic choice, the *immediate choice*:

$$IC_L(s^t) = H(S^{t+1}|s^t) = H(\pi_L^t(a|s^t)). \tag{3}$$

Immediate choice (*IC*) can be easily computed as the entropy over its policy conditioned on the current state. Even though the assumptions made for immediate choice often do not hold in complex or real-world environments, we found empirically that this objective can yield good results.

## 3.2 OPTIMALITY OF CHOICE AS AN INSTRUMENTAL CONVERGENT SUBGOAL

Turner et al. (2019) analyze the instrumental convergence of optimal agents on power-seeking subgoals and show that optimal policies tend to keep their options open (Prop. 6.9). They consider two distinct actions $a$ and $a'$ taken at a state $s'$, leading into two sets of possible future states (for an infinite horizon). These sets of future states are represented as nodes in two graphs, respectively $G$ and $G'$ (with edges weighted by the probability of transitioning from one state to another). They also assume that the states in $G \cup G'$ can only be reached from $s'$ by taking actions $a$ or $a'$. In the case where $G$ is "similar" to a subgraph of $G'$, in the sense that they are equivalent up to arbitrary swapping of pairs of states, the authors prove that the probability of $a$ being optimal is higher than the probability of $a'$ being optimal (for most reward function distributions). Therefore, if $G'$ contains more states than $G$, an optimal agent will choose $a'$ over $a$.

Turner et al. (2019) thus lend theoretical support to our proposal: while there is no guarantee that any one optimal policy (corresponding to a rational agent with arbitrary reward function) pursues higher choice, in expectation (over a bounded space of reward functions) most policies do choose actions that lead to higher choice, all else being equal. As such, while we may not know a rational agent's concrete goals, there is a high chance that choice works as an instrumental subgoal.

## 3.3 COMPARISON BETWEEN CHOICE AND EMPOWERMENT

The empowerment (Klyubin et al., 2005) of a leader agent in a given state $s^t$ and for horizon $n$ is $\mathcal{E}_L^n(s^t) = \max_{\omega(a^n|s^t)} I(S^{t+n}; A^n|s^t) = \max_{\omega(a^n|s^t)} H(S^{t+n}|s^t) - H(S^{t+n}|A^n, s^t)$, with $a^n$ as a sequence of $n$ actions of the leader agent and $\omega$ as a probing distribution over its $n$-step action sequences. When setting the probing distribution $\omega$ equal to the leader agent's policy, equation 3.3 simplifies to $\mathcal{E}_L^n(s^t) = EC_L^n(s^t) - H(S^{t+n}|A^{t+n}, s^t)$, with $EC_L^n(s^t)$ as the entropic choice of the leader agent introduced in equation 2. If we further assume deterministic environment transitions, then empowerment becomes equal to entropic choice, i.e. $\mathcal{E}_L^n(s^t) = EC_L^n(s^t)$.

In contrast to the previously introduced methods to estimate choice of another agent, empowerment of another agent cannot be estimated from observations of the environment transitions. To estimate another agent's empowerment in a given state ($\mathcal{E}_L^n(s^t)$), access to its action space as well as privileged access to an environment simulator are be required, which violates the main assumption of our research work, i.e. learning to assist others only from observations of the environment transitions. Even when assuming privileged access, computing empowerment in large or continuous-state environments often remains infeasible (Mohamed and Rezende, 2015; Gregor et al., 2016; Zhao et al., 2020), as it requires maximizing over all possible probing distributions $\omega$ of the leader agent. In contrast, estimating state entropy, as needed for the computation of the metrics introduced in this work, is feasible in large and continuous environments (Seo et al., 2021; Mutti et al., 2020).

## 3.4 BEHAVING ALTRUISTICALLY BY MAXIMIZING ANOTHER AGENT'S CHOICE

Having considered three methods to estimate an agent's choice (eq. 1-3) we now apply them to a Markov Game of two agents. The main hypothesis is that maximizing the choice of another agent is likely to allow it to reach more favourable regions of the state-space (for many possible policies of the agent), thus supporting it without a task-specific reward signal.

**Altruistic agent's policy definition.** In this Markov Game, one agent is the leader, with the subscript $L$, and another one is the altruistic agent, with the subscript $A$. We define the optimal policy of the altruistic agent as the one that maximizes the future discounted choice of the leader,

$$\pi_A^* = \operatorname*{argmax}_{\pi_A} \sum_{t=0}^{\infty} \gamma_A^t \, C_L(s^t), \tag{4}$$

where the generic choice $C_L(s^t)$ can be estimated by one of several methods: discrete choice $DC_L^n(s^t)$, entropic choice $EC_L^n(s^t)$ or immediate choice $IC_L^n(s^t)$.

**Conditional estimates of choice.** As the agents interact in the same environment, they both have influence over the system state $s$, which contains the state of both agents. This makes applying single-agent objectives based on the state distribution (such as eq. 1 and 2) difficult to translate to a multi-agent setting, since the states of both agents are intermingled. For example, an altruistic agent that maximizes entropic choice naively (eq. 2) will maximize both the state availability of the leader agent (which mirrors the single-agent entropic choice) and its own state availability (which does not contribute towards the altruism goal).

To maximize entropic choice without also increasing the entropy of the altruistic agent's actions, we propose to *condition* the choice estimate on the altruistic agent's actions over the same time horizon, denoted by the random variable $A_A^{t:t+n-1}$:

$$EC_L^n(s^t) = H(S^{t+n}|A_A^{t:t+n-1}, \pi_L, s^t). \tag{5}$$

In order to better understand eq. 5, we can use the chain rule of conditional entropy (Cover and Thomas, 2005, ch. 2) to decompose it into two terms: $EC_L^n(s^t) = H(S^{t+n}, A_A^{t:t+n-1}|\pi_L, s^t) - H(A_A^{t:t+n-1}|\pi_L, s^t)$, respectively the joint entropy of the states and actions, and the entropy of the actions. Therefore, we can interpret this objective as the altruistic agent maximizing the variety of states and actions, but subtracting the variety of its own actions, which is the undesired quantity. We can also relate eq. 5 to discrete choice (eq. 1). Using the fact that $H(X|E) \leq \log|\text{range}(P(X|E))|$ for a random variable $X$ and event $E$ (Galvin, 2014, Property 2.12), we see that eq. 5 is a lower bound for a count-based choice estimate (analogous to eq. 1), also conditioned on the altruistic agent's actions: $EC_L^n(s^t) \leq \log DC_L^n(s^t) = \log|\text{range}\left(S^{t+n}|A_A^{t:t+n-1}, \pi_L, s^t\right)|$. However, assuming simultaneous actions, the immediate choice estimate (eq. 3) stays unchanged, i.e. $IC_L(s^t) = H(\pi_L^t(a|s^t)|a_A^t) = H(\pi_L^t(a|s^t))$. The technical details of how these estimates can be computed from observations of the environment transitions are given in Appendix A.

# 4 EXPERIMENTAL EVALUATION

We introduce three multi-agent environments of increasing complexity[1], in which the success of a leader agent depends on the behaviour of one or more additional agents. In each environment, we first evaluate a subset of the proposed methods for choice estimation ($DC_L^n$, $EC_L^n$ and $IC_L$) by comparing the estimated choice of the leader agent in minimalistic scenarios. We then evaluate our approach of behaving altruistically towards others by maximizing their choice (section 3.4) and measure performance of our approach as the reward achieved by the leader agent. We provide videos of the emergent behaviours in the supp. mat. (see appendix F). We compare our method to both an unsupervised and a supervised approach. Note that the supervised approach has stronger assumptions, as it requires direct access to the leader agent's reward function. We do not consider inverse RL (IRL) as a relevant baseline, as it would rely on demonstrations of expert behaviour, which we do not assume. Even if perfect knowledge of the state transition probabilities is assumed, this does not allow generating expert demonstrations of the leader agent's policy, as its expert policy would in turn depend on the policy of the altruistic agent, which is yet to be found by IRL.

## 4.1 DISCRETE ENVIRONMENTS WITH CONTROLLABLE GATES

We start by considering three different scenarios on a grid, illustrated in Fig. 1 (top row), with the starting positions of the leader (green) and an additional agent (blue) shown in faded colors, obstacles are gray, and agents may move in one of the four cardinal directions or stay still.

**Choice estimate analysis.** We first verify whether the estimated choice for each state (agent position) correctly maps to our intuitive understanding of choice (that is, the diversity of actions that can be taken). Therefore, we conducted an analysis of the estimated choice of the leader agent using a simplified version of the environment (Fig. 1, top left), in which only the leader agent is present and selects actions uniformly at random. Fig. 1 (bottom row) shows the three different methods of estimating choice evaluated for each possible cell position of the leader agent. We can observe that states in less confined areas, e.g. further away from walls, generally feature higher choice estimates, with the least choice being afforded by the dead end at the right. All three method's estimates are qualitatively similar, which validates the chosen approximations. In line

---

[1]In appendix E, we evaluate performance in a non-spatial environment.

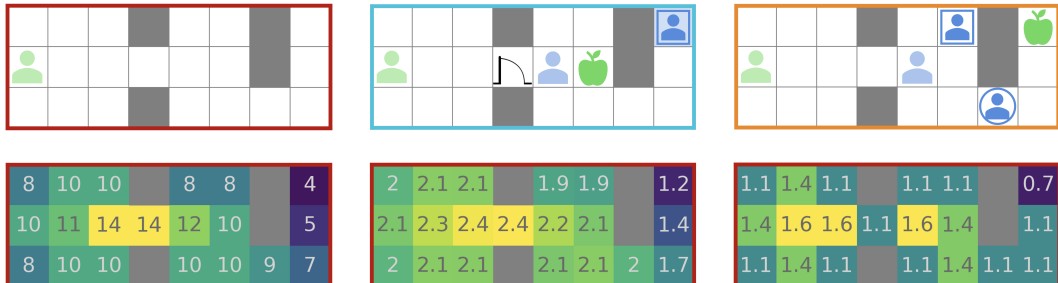

Figure 1: *Top row:* Single-occupancy grid environments in which agents can either move to a free adjacent cell or stay still. The green apple (reward +1) can only be obtained by the leader agent (in green) and no other external rewards exist. Grey cells are blocked. *Bottom row:* Visualisation of the estimated choice of the leader agent when positioned at the respective cells. Left to right: discrete choice $DC_L^3$, entropic $EC_L^3$ and immediate choice $IC_L$ (eq. 1, 2 and 3).

with the simplifications made, the immediate choice (IC) estimates tend to be more local, as can be observed when comparing the estimates for the cell at row 2, column 4. In conclusion, these results qualitatively agree with an intuitive understanding of choice of an agent in a grid environment.

**Environment setup.** In the Door Scenario (Fig. 1, top center), the door switch (row 1, col. 8) can only be operated by the altruistic agent. The door (row 2, col. 4) remains open as long as the altruistic agent is on the switch cell and is closed otherwise. As the leader agent always starts to the left of the door and the altruistic agent to the right, the leader agent can only attain its goal, the apple (row 2, col. 6), if the altruistic agent uses the door switch to enable the leader agent to pass through the door. In the Dead End Scenario (Fig. 1, top right), the door is always open, and the leader agent's target object (green apple) is moved to the top right cell. Hence, the leader agent can obtain the apple without additional help from the altruistic agent. However, the altruistic agent could potentially block the path by positioning itself at the entry to the dead end. This situation would be the opposite of altruistic behaviour and is, of course, undesired. We compare to a supervised approach, to Assistance via Empowerment (AvE, (Du et al., 2020)) and a random-policy baseline.

**Assistance via Empowerment baseline.** We compare with the recently-proposed AvE, which has a similar goal (Du et al., 2020). There are two major differences: AvE is not unsupervised, and it requires privileged access to an environment simulator to produce estimates. Hence, its use in real or black-box environments is limited. We used the authors' implementation with fixed hyper-parameters, except for the crucial horizon $n$, for which we present a sweep in app. B.

**Training.** We start by pretraining the leader agent with Q-Learning (Watkins and Dayan, 1992), with the altruistic agent executing a random policy. Hence, after convergence, the leader agent's policy targets the green apple. Appendix B lists all details and parameters. Afterwards, the leader agent's learning is frozen and the altruistic agent is trained; it always observes the position of the leader agent $s_L$, its own position $s_A$, and the environment state $s_{\text{env}}$, which is composed of the door state (open, closed) and the food state (present, eaten). The altruistic agent is trained with Q-Learning to maximize the discounted future choice of the leader agent (see eq.. 4. For that, it uses one of the three proposed methods such as eq. 3, eq. 2 or eq. 1, as detailed in appendix A.1.

**Results.** We investigate the developed behaviour of the altruistic agent after convergence for different choices of the hyperparameters – look-ahead horizon $n \in \{1, 3, 12\}$ (which determines the scale at which choices are considered) and discount factor $\gamma_a \in \{0.1, 0.7\}$ (which defines whether the altruistic agent gives higher importance to the short-term or long-term choice of the leader agent). Success is binary: either the leader agent attains its goal (green apple), or not.

In the Door Scenario (Fig. 1, top center), we found that, for longer horizons $n$ and higher discount factors $\gamma_a$, the altruistic agent opens the door to allow the leader agent to reach its target, by occupying the switch position (square outline; row 1, col. 8). For smaller $n$ and lower $\gamma_a$, the altruistic agent does not execute any coordinated policy and the leader does not succeed. Using the AvE method, we find that it only opens the door for $n = 3$, but fails to do so for $n = 1$ and $n = 12$.

In the Dead End Scenario (Fig. 1, top right), we observe that, for longer horizons $n$ and large discount factors $\gamma_a$, the altruistic agent stays out of the leader agent's way by occupying a far-away cell (square outline; row 1, col. 6). For short horizons $n$ and high discount factors $\gamma_a$, the altruistic

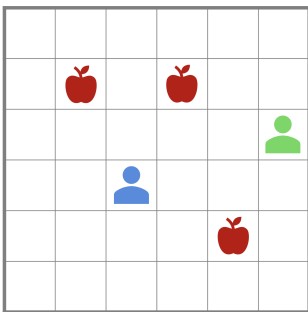 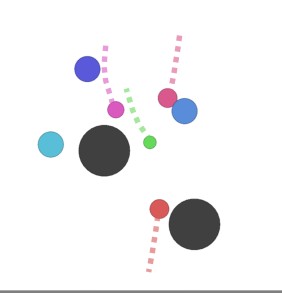

Figure 2: *Left*: In Level Based Foraging (LBF) two agents must cooperate to receive rewards for eating apples. The leader agent is green, altruistic agent is blue. *Right*: In Tag, a leader (green) tries to escape from adversaries (red colors). Altruistic agents (who may help the leader) are blue.

Table 1: Comparison of the estimated choice (IC) of a leader agent in a high performance scenario (with a cooperative partner) vs. a low performance scenario (with a randomly-acting partner). IC correlates with high performance.

|  |  | Performance | |
|---|---|---|---|
|  |  | Low | High |
| Tag | IC | 26.9% | 56.7% |
|  | Reward | -7.55 | -2.11 |
| LBF | IC | 19.1% | 55.3% |
|  | Norm. Reward | 4.0% | 94.3% |

agent actively blocks the entry to the hallway that contains the target (circle outline; row 3, col. 7), to prohibit the leader agent from entering this region of low estimated choice (recall that the choice for each cell is visualized in Fig. 1, bottom right). This failure case can be prevented by having a large enough horizon $n$ and discount factor $\gamma_a$, analogously to the selection of the temperature hyperparameter in maximum entropy single-agent RL (Haarnoja and Abbeel, 2018). We find that this configuration performs consistently better than others in both scenarios, and hence is more preferred. On the other hand, the AvE method does not block the path of the leader agent for $n = 1$, but blocks its path for $n = 3$ and $n = 12$.

We found that the resulting behaviour of our approach is independent of the used method for choice estimation, i.e. either discrete choice (eq. 1) or entropic choice (eq. 2) yield the same outcome, with immediate choice (eq. 3) being a special case of entropic choice. As for the AvE baseline, we hypothesize that the variance of results is due to the nature of the proxy used in practice, which includes components of empowerment from both agents (sec. 3.4). The binary outcomes for all hyperparameter combinations are given in appendix B. We also compare to a supervised baseline (receiving a reward when the leader obtains the apple), in which case the leader always succeeds.

## 4.2 LEVEL-BASED FORAGING EXPERIMENTS

**Computational efficiency.** Due to the computational complexity resulting from the need to estimate a long-term distribution of states, $p(s^{t+n}|s^t)$, we focus on immediate choice (IC) to estimate the leader agent's choice in the remaining sections. Furthermore, in rare state-action sequences, the assumptions made for IC, i.e. deterministic environment transitions and an injective relationship from actions to states, may not hold. Nonetheless, we did not find this to adversely affect the results. Due to its dependence on access to the environment simulator and its computational complexity, we do not consider the AvE baseline for the remainder of experiments.

**Setup.** We use a fully-observable multi-agent environment that enables us to assess the level of cooperation among agents (level-based foraging, LBF, Christianos et al. (2020)) to evaluate the performance of altruistic agents in more complex environments with discrete state spaces. We compare our method to a maximum-entropy approach from single-agent RL (Mutti et al., 2020) and a random-policy baseline. A visualization of the environment is depicted in Fig. 2 (left). The two agents can forage apples by simultaneously taking positions at different sides of a targeted apple, yielding a fixed reward. We first train two agents – which receive an equal reward for foraging – using Deep Q-Learning (DQL, Van Hasselt et al. (2015)), corresponding to fully-supervised shared-reward in multi-agent reinforcement learning (MARL). We then take one of these pretrained agents that has learned to forage apples when accompanied by a cooperating agent, freeze its policy, and place it as the leader agent (green) into the test scenario (additional details are provided in app. C).

**Choice estimate analysis.** We first qualitatively evaluate IC as an estimator for choice in Fig. 3, by comparing representative scenarios. To quantitatively analyse IC as an estimator for the leader agent's choice, we compare the leader agent's average IC (over 100 episodes) in two scenarios, one in which it can acquire many rewards, i.e. the other agent acts cooperatively, and one where it can

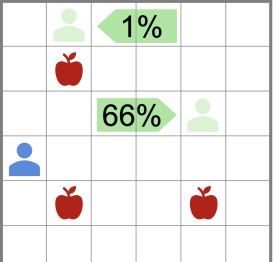 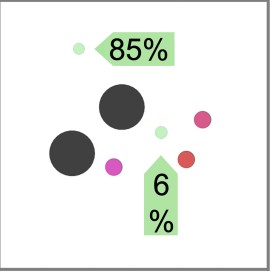 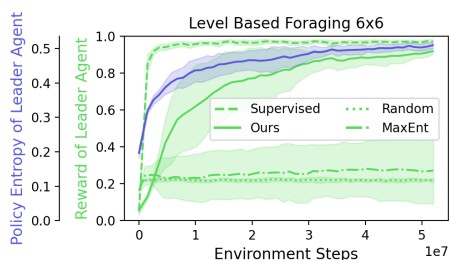

Figure 3: For two representative positions of the leader agent, its immediate choice (IC) estimates are given as a percentage of the maximum possible. **Left (LBF):** The leader agent has low IC when having to wait for another agent to support it in foraging an apple and high IC when having the option of approaching multiple apples. **Right (Tag):** The leader has low IC when chased by the adversaries and high IC when it has multiple escape paths.

Figure 4: Normalized reward of the leader agent (right vert. axis, green) when the altruistic agent is trained to maximize the leader's choice (ours), acting (random), receiving the same reward as the leader (supervised), or maximizing the state entropy (Mutti et al., 2020). Left vert. axis (blue): internal reward of our altruistic agent (estimated choice of the leader).

acquire only few rewards, i.e. the other agent takes random actions. We show the results in Table 1. We observe that the leader agent's estimated choice is substantially higher when it is able to acquire high rewards. Note that the IC estimate does not have privileged access to the reward function of the leader agent, and so this experiment evaluates its worth as a generic proxy for the leader's reward. Assuming that an agent is able to acquire higher rewards when having more choice, these results indicate that IC is a reasonable estimator for the leader agent's choice in LBF.

**Training.** We now consider an environment that consists of the previously pretrained leader and an additional altruistic agent, which is trained from scratch and does not receive a reward for foraging apples, but is rewarded according to the leader agent's choice. Its reward is given as the current estimate of the leader agent's IC (eq. 3) and it is trained using DQL. To compute its internal reward signal, the altruistic agent would therefore need to estimate the state transition probabilities, as detailed in A.2. To decouple our approach's performance from that of the state transition estimator, we instead directly compute the altruistic agent's reward using the leader agent's policy.

**Results.** We define the performance of the altruistic agent not as its achieved internal reward but as the reward achieved by the leader agent, i.e. its performance in enabling the leader agent to forage apples. Fig. 4 shows a comparison of the altruistic agent's performance to that achieved by 3 baselines (two unsupervised and one supervised), averaged over 5 random seeds, with the standard deviation as the shaded area. It can be observed that the performance of the altruistic agent converges to a similar performance to that of the supervised agent, and outperforms the baseline approaches by a large margin. Furthermore, the IC improvement of the leader agent is correlated with its reward improvement, which supports using IC as a reasonable proxy for the choice of the leader agent.

### 4.3 MULTI-AGENT TAG GAME WITH PROTECTIVE AGENTS

**Setup.** We use a multi-agent tag environment (Tag, Mordatch and Abbeel (2018); Lowe et al. (2017); Terry et al. (2020)), illustrated in Fig. 2 (right), to evaluate the capabilities of altruistic agents in complex environments with continuous state spaces. Adversaries are rewarded for catching the leader, which in turn receives a negative reward for being caught or crossing the environment boundaries. To speed up training, altruistic agents additionally receive a small negative reward for violating the environment boundaries. We pretrain the adversaries and the leader (without the presence of altruistic agents) using MADDPG (Lowe et al., 2017) and DDPG (Lillicrap et al., 2016) respectively. After pretraining, the adversary agents have learned to cooperatively chase the leader agent, which in turn has learned to flee from the adversaries. Exact setup specifications and all parameters are given in appendix D.

**Choice estimate analysis.** As done for LBF, we evaluate the IC of the leader agent in representative scenarios in Fig. 3. We also quantitatively evaluate IC as an estimator for the leader agent's choice, by comparing the leader agent's IC per timestep for a scenario in which it receives high rewards to

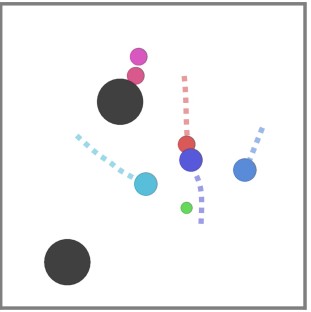 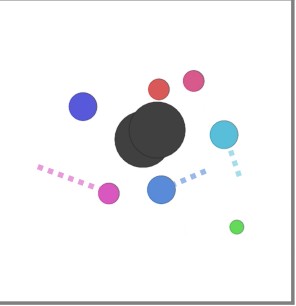

Figure 5: Example behaviour of altruistic agents (blue) that learned to actively defend the leader agent (green) from the adversaries (red) in Tag. Obstacles are black. The trajectories taken by some of the agents in the last 10 steps are shown as dotted lines.

Table 2: Results of the Tag experiments (mean and standard deviation over 5 random seeds). Refer to sec. 4.3 for more details.

| Method | # Caught $\downarrow$ |
|---|---|
| None | 7.55±0.56 |
| Random | 7.12±1.08 |
| Cage | 6.80±1.12 |
| Supervised | 6.94±1.13 |
| Supervised + Cage | 6.80±1.37 |
| **Ours** | **2.87±0.96** |

one where it receives low rewards. We again hypothesize that the leader agent is able to acquire higher rewards when having more choice. Table 1 shows that the estimated choice is substantially higher in the high-success scenario, indicating that IC is a reasonable estimator also in Tag.

**Training.** We freeze the pretrained policies of the adversary agents and the leader agent and insert three additional altruistic agents which observe all agents but are not observed themselves. Each additional altruistic agent's internal reward signal is given as the IC of the leader agent (equation 3), which is directly computed as done in LBF (see 4.2).

**Results.** Performance of the altruistic agents is defined as the times per episode that the leader agent is caught by the adversaries, i.e. the lower the better. In Table 2, the performance of the team of three altruistically trained agents (ours) is compared to three relevant baselines, with the altruistic agents either removed (None), acting randomly (random), or solely receiving a small negative reward for violating the environment boundaries (cage). In contrast to LBF, we do not compare to an unsupervised exploration approach, as we are not aware of such an implementation for cooperative MARL. Additionally, we report results for the case in which the altruistic agents receive the same reward as the leader agent (supervised), possibly appended by a negative reward for violating the environment boundaries (supervised + cage). It can be observed that our approach outperforms all relevant baselines by a substantial margin and also outperforms the supervised approach. We hypothesize this to be due to the dense internal reward signal that our approach provides, as compared to the sparse rewards in the supervised scenario: recall that in the supervised scenario the additional altruistic agents receive a large negative reward only when the leader agent is caught by the adversaries, whereas our approach provides a dense reward signal that corresponds to the current estimate of the leader agent's choice. Fig. 5 displays the emerging protective behaviour of altruistic agents trained with our approach. Results videos are found in the supplemental material.

## 5 CONCLUSIONS

We lay out some initial steps into developing artificial agents that learn altruistic behaviour from observations and interactions with other agents. Our experimental results demonstrate that artificial agents can behave altruistically towards other agents without knowledge of their objective or any external supervision, by actively maximizing their choice. This objective is justified by theoretical work on instrumental convergence, which shows that for a large proportion of rational agents this will be a useful subgoal, and thus can be leveraged to design generally altruistic agents.

This work was motivated by a desire to address the potential negative outcomes of deploying agents that are oblivious to the values and objectives of others into the real world. As such, we hope that our work serves both as a baseline and facilitator for future research into value alignment in simulation settings, and as a complementary objective to standard RL that biases the behaviour towards more altruistic policies. In addition to the positive impacts of deployed altruistic agents outside of simulation, we remark that altruistic proxy objectives do not yet come with strict guarantees of optimizing for other agents' rewards, and identify failure modes (sec. 4.1) which are hyperparameter-dependent, and which we hope provide interesting starting points for future work.

## 6 ETHICS STATEMENT

We addressed the relevant aspects in our conclusion and have no conflicts of interest to declare.

## 7 REPRODUCIBILITY STATEMENT

We provide detailed descriptions of our experiments in the appendix and list all relevant parameters in table 4. All experiments were run on single cores of Intel Xeon E7-8867v3 processors (2.5 GHz). Training times are given in the respective sections in the appendix. For the LBF and Tag experiments, we report mean and standard deviation over five different random seeds. The Gridworld experiments yield deterministic results. We will provide the source code for all experiments conducted with the final version of this publication. We created detailed instructions on how to run the code in order to replicate the experimental outcomes presented in this work.

## 8 ACKNOWLEDGEMENTS

We thank Thore Graepel and Yoram Bachrach for their helpful feedback. We are also grateful to the anonymous reviewers for their valuable suggestions. This work was supported by the Royal Academy of Engineering (RF\201819\18\163).

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

## A    ESTIMATION OF LEADER AGENT'S CHOICE FROM OBSERVATION

### A.1    MODEL-BASED ESTIMATION OF CHOICE FROM OBSERVATIONS

We introduce a model-based estimator of choice that is suitable for small-scale discrete-state environments, having the advantage that it is easily interpretable. Recalling how we compute the *discrete choice* and *entropic choice* estimates for the leader agent, an estimate of the $n$-step state distribution conditioned on the altruistic agent's actions is needed, i.e. $P(s^{t+n}|\pi_L, a_A^{t:t+n-1}, s^t)$. To simplify this computation, we assume the altruistic agent's action to equal *hold* for the next $n$ steps. More specifically, we assume that the altruistic agent's state is unchanged for the next $n$ steps. Furthermore assuming that both the state and the action space are discrete, we compute

$$P(s^{t+n}|\pi_L, a_A^{t:t+n-1}, s^t) = s^t\, T(s_A^t)^n, \tag{6}$$

with

$$T(s_A^t)_{ij} = P(s^{t+1} = s_j \mid s^t = s_i, s_A^{t+1} = s_A^t) \tag{7}$$

where the state transition matrix $T(s_A)$ holds the transition probabilities between all possible states, as a function of the state of the altruistic agent $s_A$. To compute $T(s_A)$, the system state is encoded into a one-hot vector $s_{\mathbb{1}}$.

The $n$-step *discrete choice* of the leader agent can then be computed as

$$DC_L^n(s^t) = \|s_{\mathbb{1}}^t\, T(s_A^t)^n\|_0, \tag{8}$$

its $n$-step *entropic choice* as

$$EC_L^n(s^t) = H\big(s_{\mathbb{1}}^t\, T(s_A^t)^n\big), \tag{9}$$

and its *immediate choice* as

$$IC_L(s^t) = H\big(\pi_L^t(a|s^t)\big) = H\big(s_{\mathbb{1}}\, T(s_A^t)\big) \tag{10}$$

In environments with a discrete state and action space, the altruistic agent can hence use an estimate of the state transition matrix $T$ to estimate the choice of the leader agent using either of the proposed methods, i.e. *DC*, *EC* or *IC*. An estimate of $T$ can be built over time, by observing the environment transitions and computing the transition probabilities as relative frequencies of observed transitions.

### A.2    MODEL-FREE ESTIMATION OF CHOICE FROM OBSERVATIONS

To limit the computational complexity, which is important for environments with large or continuous state spaces, we also consider *immediate choice* as an estimator for the leader agent's choice $(IC_L(s^t) = H(S^{t+1}|s^t))$. As shown in section 3.1, this estimate can be simplified to $H(S^{t+1}|s^t)) = H(\pi_L^t(a|s^t))$, under the named assumptions. Hence, to compute the immediate choice of the leader, the altruistic agent requires an estimate of the leader agent's policy entropy, which can be learned from observation using a policy estimation network (Hong et al., 2018; Papoudakis et al., 2020; Mao et al., 2019; Grover et al., 2018).

## B    GRIDWORLD EXPERIMENTS

### B.1    TRAINING PROCEDURE

#### B.1.1    SETUP

The environment setup is described and displayed in section 4.1.

**AvE baseline.**    We evaluate the AvE baseline for different horizons $n$. For each horizon, we tested the AvE baseline as implemented in the provided source code[2], using the hyper-parameters suggested by the authors. The original implementation uses a look-ahead horizon $n = 10$. We found

---

[2]https://github.com/yuqingd/ave

Table 3: Results on minimalistic environments (sec. 4.1). Two desired behaviours of the altruistic agent, one for each scenario, are listed as the two bottom rows. In the columns, $D$ denotes Discrete Choice, $E$ Entropic Choice, AvE the Assistance via Empowerment baseline (Du et al., 2020), and SV the supervised baseline. Refer to sec. 4.1 for a discussion of the results.

| | | $n = 1$ | | | | | $n = 3$ | | | | | $n = 12$ | | | | | |
| | | $\gamma_a = 0.1$ | | $\gamma_a = 0.7$ | | | $\gamma_a = 0.1$ | | $\gamma_a = 0.7$ | | | $\gamma_a = 0.1$ | | $\gamma_a = 0.7$ | | | |
| Category | Scen. | $D$ | $E$ | $D$ | $E$ | AvE | $D$ | $E$ | $D$ | $E$ | AvE | $D$ | $E$ | $D$ | $E$ | AvE | SV |
|---|---|---|---|---|---|---|---|---|---|---|---|---|---|---|---|---|---|
| Opens door | A | ✗ | ✗ | ✗ | ✗ | ✗ | ✗ | ✗ | ✓ | ✓ | ✓ | ✓ | ✓ | ✓ | ✓ | ✗ | ✓ |
| Non blocking | B | ✓ | ✓ | ✗ | ✗ | ✓ | ✓ | ✓ | ✓ | ✓ | ✗ | ✓ | ✓ | ✓ | ✓ | ✗ | ✓ |

that results are equal for both $n = 10$ and $n = 12$, which is why we only display results for $n = 12$. We further evaluated the AvE baseline for $n$ between 1 and 12. For the *Opens door* task, we found that AvE yields success for $n = 2, 3, 4, 5$ and failing for the remaining. For the *Non blocking* task, we found that AvE yields success for $n = 1, 2$ and failing for the remaining.

### B.1.2 PRETRAINING

We first pretrain the leader agent using tabular Q-Learning, with learning parameters given in Table 4. During this pretraining, the altruistic agent takes random actions. We train until all Q-Values are fully converged, i.e. training runs for 300000 environment steps.

### B.1.3 REWARD COMPUTATION FOR ALTRUISTIC AGENTS

The altruistic agent is then also trained using tabular Q-Learning, and its internal reward signal is given as the choice estimate of the leader agent, i.e. either $DC_L^n(s^t)$, $EC_L^n(s^t)$ or $IC_L(s^t)$, which is computed with the model based-estimation introduced in appendix A.1. The altruistic agent records all environment transitions and frequently updates its estimate of the state transition matrix $T(s_A)$, which is needed to compute the internal reward signal for the altruistic agent. All training parameters can be found in Table 4. Training time is about 15 minutes per experiment.

## B.2 PERFORMANCE EVALUATION

Performance of the altruistic agent is reported for two different categories, as shown in Table 3. For each category, we report success or failure for choice estimate look-ahead horizons $n \in \{1, 3, 12\}$ and discount factors of the altruistic agent $\gamma_a \in \{0.1, 0.7\}$. Success or failure was always deterministic, conditioned on the experiment setup, i.e. 10 simulations were run for each setup which always yielded the same outcome. To estimate the leader agent's choice, the altruistic agent uses either *discrete choice* ($D$, equations 1 and 8) or *entropic choice* ($E$, equations 2 and 9). It must be noted that horizon $n = 12$ is equivalent to an infinite horizon look-ahead for the given environment size and that *entropic choice* is equivalent to *immediate choice* (equations 3 and 10) at horizon $n = 1$, as the environment satisfies the necessary conditions listed for equation 3.

Table 3 displays the results of this experiment. In the first row, it is evaluated whether the altruistic agent opens the door at all times, such that the leader agent can eat the green apple. It can be observed that the altruistic agent only opens the door for longer horizons $n$, respectively higher discount factors $\gamma_a$.

Given the definitions of *discrete choice* (Equation 1) and *entropic choice* (Equation 2), it can be assumed that the choice horizon $n$ determines the locality for which choice is considered and that the discount factor $\gamma_a$ defines whether the altruistic agent gives higher importance to the short-term or long-term choice of the leader agent. This is in line with the observed results for the first category (Opens door). It can be assumed that, for short horizons $n$, the altruistic agent does not open the door, as it does not estimate that this would lead to an increase in the leader agent's choice. A similar argumentation follows for low discount factors $\gamma_a$.

The bottom-row category evaluates whether the altruistic agent does not block the hallway that leads up to the leader agent's target apple in the top right environment cell. This category demonstrates a possible failure case of the proposed approach of maximizing another agent's choice. For short horizons $n$ and high discount factors $\gamma_a$, the altruistic agent actively blocks the entry to the low-entropy hallway towards the top right cell – by constantly occupying cell $(2, 6)$ – to prohibit the

leader agent from entering this region of low estimated choice. This failure case can be prevented by an appropriate selection of the hyperparameters – horizon $n$ and discount factor $\gamma_a$. It is related to the selection of the temperature hyperparameter in maximum entropy single-agent RL (Haarnoja and Abbeel, 2018); if chosen incorrectly, the agent does not foster environment rewards in low-entropy regions. A possible solution to this problem would be to define a constrained optimization problem, as shown by Haarnoja and Abbeel (2018).

### B.3 Ablation study on joint learning

**Training.** To investigate the effects of joint learning of the leader agent's and the altruistic agent's policy, we adapted the training process described in section 4.1 for the Gridworld experiments as following. Instead of first learning the policy of the leader agent while the altruistic agent takes random actions, we initialized both policies from scratch and trained both agents simultaneously with the parameters given in Table 4.

**Results.** We evaluated the outcome for the same scenarios, i.e the scenarios described in section 4.1. We found that the results for the individual test cases were equivalent to those achieved when training the leader and the altruistic agent subsequently, i.e. the results are equivalent to those displayed in Table 3.

## C    Level Based Foraging experiments

### C.1    Training procedure

#### C.1.1    Setup

We adopted the Level Based Foraging[3] environment as given in Christianos et al. (2020). We only focus on two-agent scenarios and only consider the subset of possible environments that require full cooperation among agents, i.e. those where food can only be foraged by two agents cooperatively. We therefore only consider environments where both agents are at level one, and all present food is at level two. In the original implementation, both agents have to simultaneously select the $eat$ action while docking at different sides of a food object to forage the object and receive the reward. To reduce training time, we simplify this setup by reducing the action space to $up, down, left, right, stay$, i.e. we remove the $eat$ action and enable agents to forage food by being simultaneously at different sides of a food object, with no further action required.

#### C.1.2    Pretraining

To obtain a pretrained leader agent, we first train two agents in the environment that are equally rewarded for foraging food. This setup corresponds to shared-reward cooperative MARL (Tan, 1993). Both agents are trained using Deep Q Learning (DQL, (Van Hasselt et al., 2015)), using a fully connected neural network with two hidden layers and five output values, resembling the Q values of the five possible actions. The exact training parameters are listed in Table 4. We then take either one of the two agents and set it as the pretrained leader agent for the subsequent evaluation of the altruistic agent.

#### C.1.3    Training of additional agents

We then insert an additional agent into the environment that shall act altruistically towards the leader agent. This additional agent is trained in the same fashion and with the same parameters as the previously trained leader agents. Only its reward signal is different, as laid out in the next section.

#### C.1.4    Reward computation for additional agents

We compare four different approaches for how the reward of the additional agent is defined, respectively how it behaves. *Random:* The agent takes random actions. *Supervised:* The agent receives the same reward as the leader agent, i.e. a shared reward as in cooperative MARL. *Ours:*

---

[3]https://github.com/semitable/lb-foraging

The reward of the additional agent is defined as the *immediate choice* of the leader agent, as detailed in equation 3. We compute the leader agent's policy entropy by computing the entropy of the softmax of the leader agent's Q values in the given state. We further consider an *unsupervised* baseline, as detailed in the next paragraph.

**Unsupervised baseline (MaxEnt).** As an unsupervised baseline, we implemented the MEPOL approach of Mutti et al. (2020). Their task-agnostic unsupervised exploration approach maximizes the entropy over the state distribution of trajectory rollouts. For this baseline, the additional agent is trained with the implementation given by the authors[4], which itself builds on TRPO (Schulman et al., 2015). We leave all parameters unchanged but evaluate different learning rates; $lr \in \{1e - 6, 1e - 5, 1e - 4, 1e - 3, 1e - 2, 1e - 1\}$. Best results were achieved for a learning rate of $1e - 5$, which was hence picked as the relevant baseline.

## C.2 Performance evaluation

Each experiment was run for 5 different random seeds and mean and standard deviation are reported. Training progress is shown in Figure 4. Evaluations are computed every 10000 environment steps for 200 episodes, with the exploration set to zero. Training time was about 14 hours for each run. Results are shown in Fig. 4.

## D Tag experiments

### D.1 Training procedure

#### D.1.1 Pretraining

We use the Simple Tag (Tag) implementation by Terry et al. (2020)[5] which is unchanged as compared to the original implementation of Mordatch and Abbeel (2018)[6], only fixing minor errors. We first adopt the original configuration and pretrain three adversaries and one good agent (leader agent) using the parameters listed in Table 4. We use MADDPG (Lowe et al., 2017)[7] to train adversary agents, and modify the framework as follows. The last layer of each agent's actor-network outputs one value for each of the environment's five possible actions, over which the softmax is computed. We then sample the agent's action from the output softmax vector, which corresponds to the probabilities with which the agent takes a specific action in a given state. We train the leader agent with DDPG (Lillicrap et al., 2016),[7] where we equally modify the output layer. Each actor and critic network is implemented as a fully-connected neural network with two hidden layers, with layer sizes as given in Table 4.

To make the environment more challenging for the leader agent, we decrease its maximum speed and acceleration to $70\%$ of the original value. We next insert three additional agents into the environment whose observations include all agents and objects. These additional agents are not observed by adversary agents or the leader agent. The additional agents are of the same size as the adversary agents, and their acceleration and maximum velocity are equal to that of the leader agent. To speed up training, we made the following changes to the environment, which are applied to our approach as well as to all baselines. First, we spawn the three additional agents in the vicinity of the leader agent, which itself is spawned at a random position. Furthermore, we randomly pick two out of the three adversary agents and decrease their maximum acceleration and maximum speed by 50%. We made these changes to be able to observe substantial differences between the different approaches after a training time of less than 24h.

#### D.1.2 Training of additional agents

We train these three additionally inserted agents with the previously described modified version of MADDPG. The reward for each agent is defined either according to our developed approach, or any of the given baselines, as detailed in the next section.

---

[4] `https://github.com/muttimirco/mepol`
[5] `https://github.com/PettingZoo-Team/PettingZoo`
[6] `https://github.com/openai/multiagent-particle-envs`
[7] `https://github.com/starry-sky6688/MADDPG`

Table 4: Detailed hyper-parameters used in the three experimental environments.

| | Gridworld | Tag | Level Based Foraging | Resources |
|---|---|---|---|---|
| Environment Steps | 300000 | 7500000 | 50000000 | 300000 |
| Episode Length | 25 | 25 | 10/15 | 10 |
| Learning Rate Actor | - | 0.001 | - | - |
| Learning Rate Critic/ Q-Learning | 0.01 | 0.001 | 0.001 | 0.005 |
| Exploration Noise | 0 | 0.1 | 0 | 0 |
| Epsilon $\epsilon$ start | 0.1 | - | 1.0 | 0.1 |
| Epsilon $\epsilon$ final | 0.1 | - | 0.2 | 0.1 |
| Discount Factor $\gamma$ | 0.9 | 0.95 | 0.9 | 0.95 |
| Target Network Update Rate $\tau$ | - | 0.01 | 0.001 | - |
| Replay Buffer Size | - | 1000000 | 200000 | - |
| Training Batch Size | - | 256 | 256 | - |
| Train every $n$ steps | - | 64 | 32 | - |
| Layer Size Adversary Agent | - | 64 | - | - |
| Layer Size Leader Agent | - | 64 | 64 | - |
| Layer Size Altruistic Agent | - | 128 | 64 | - |
| Optimizer | - | Adam | Adam | - |
| Gradient Norm Clip | - | - | 0.5 | - |
| Activation Function | - | relu | relu | - |

### D.1.3 REWARD COMPUTATION FOR ADDITIONAL AGENTS FOR DIFFERENT BASELINES

We consider the following implementations for the reward computation of the additional agents, respectively different environment configurations.

*None:* For this scenario, the additional agents are removed from the environment.

The remaining approaches purely differ in the way that the reward of the additional agents is computed. No other changes are made.

*Random:* The additional agents take random actions.

*Cage:* The additional agents receive a negative reward for violating the environment boundaries, which is equal to the negative reward that the leader agent receives for itself violating the environment boundaries (part of the original Tag implementation).

*Supervised:* The additional agents receive the same reward as the leader agent. That is, they receive a reward of -10 if the leader agent is caught by the adversaries and a small negative reward if the leader agent violates the environment boundaries.

*Supervised + Cage:* The additional agents receive the same reward as the leader agent, and an additional small negative reward if they themselves violate the environment boundaries.

*Ours:* The reward of the additional agents is defined as the *immediate choice* of the leader agent, as detailed in eq. 3. To reduce the variance in the estimate of the leader agent's immediate choice, we implement an ensemble of five pretrained actor-networks for the leader agent, evaluate the policy entropy of each network, and take the median of the achieved values as the reward for the altruistic agents. Furthermore, the additional agents receive a small negative reward for themselves violating the environment boundaries.

### D.2 PERFORMANCE EVALUATION

We train *Cage, Supervised, Supervised + Cage and Ours* for five different random seeds with parameters as detailed in Table 4. We then compute the results listed in Table 2 by freezing all weights across all networks, setting the exploration noise to zero and computing the average and standard deviation over 500 rollout episodes.

Table 5: Results on the resource environment. We evaluate two baselines (random and supervised), and four different implementations of choice. Performance is measured as the average number of consume actions that failed, over all time steps and both agents (so it takes values between 0 and 1).

|  | rand. | superv. | $n = 1$ | | $n = 3$ | |
|---|---|---|---|---|---|---|
|  |  |  | $EC(=IC)$ | $DC$ | $EC$ | $DC$ |
| avg. failures ↓ | $0.336 \pm 0.312$ | $0.081 \pm 0.185$ | $0.079 \pm 0.183$ | $0.082 \pm 0.182$ | $0.079 \pm 0.185$ | $0.071 \pm 0.181$ |

## E  RESOURCE ENVIRONMENT

### E.0.1  MOTIVATION AND OVERVIEW

This environment is a special case of the general resource-based MDP proposed by Benson-Tilsen and Soares (2016), which they used to show that intelligent agents pursue instrumentally useful subgoals. The motivation behind the choice for this environment is to evaluate our proposal in non-spatial and non-navigation environments.

In the environment, there are 3 resource types, which two "consumer" agents may consume as an action. Each consumer has different preferences (reward function), and so will only consume 2 of the resource types. A third, altruistic agent, receives one resource unit of each type to distribute among the consumers, and its goal is to satisfy the preferences of the consumers without knowing their reward function. We define its performance as the average number of times that the consumers fail to consume their preferred resource (so lower is better). We compare our method to a supervised agent that is explicitly trained with the consumers' reward function, as well as to an agent that assigns the resources randomly.

### E.0.2  ENVIRONMENT DESCRIPTION

The environment is expressed as a Markov Game (see section 3). The Markov game is composed of two human-inspired consumers with subscript $C_1$, $C_2$ and an altruistic agent with subscript $A$. Three types of resources exist, $R_X$, $R_Y$ and $R_Z$. The environment state $s$ is given by the number of resources of each type available to each of the consumers. For example, $s = [(1, 0, 1), (0, 1, 0)]$ means that agent $C_1$ has one resource each of type X and Y available, while agent $C_2$ only has one resource of type Z available. At the beginning of each time step, the altruistic agent is provided with one resource per category, i.e. $R_X$, $R_Y$ and $R_Z$. The altruistic agent can assign each resource individually to any agent or discard the resource. The altruistic agent's action space is hence defined by one sub-action per resource, i.e. $a_A = (a_A^X, a_A^Y, a_A^Z)$. Each sub-action assigns the resource either to one of the consumers or discards it. The resources are then distributed according to the action taken by the altruistic agent and the environment state is updated. Resources cannot be stacked, which means that each agent can only have one resource per category available at a time. Next, the consumers attempt to consume one resource each, according to their preference. Agent $C_1$ dislikes resource $R_Z$, hence it chooses $R_X$ or $R_Y$ with equal probability. Agent $C_2$ dislikes resource $R_X$, hence it chooses $R_Y$ or $R_Z$ with equal probability. The actions of agents $C_1$ and $C_2$ are sampled accordingly and the environment state is updated. For each round, we record how many agents failed to consume a resource that was not available.

### E.1  TRAINING

The altruistic agent is trained with Q-Learning (Watkins and Dayan, 1992) to maximize the discounted future choice of the consumers (see eq. 4). For that, it uses one of the three proposed objectives, namely IC (eq. 3), EC (eq. 2) or DC (eq. 1), which it estimates as detailed in appendix A.1. The exact hyper-parameters are given in Table 4. We compare the performance of the altruistic agent that maximizes the choice of the consumers to that of a supervised agent. The reward of the supervised agent is the negative of the number of consumers that attempted to consume a resource, in that time step, and failed. Further, we compare to a random-policy baseline that distributes the resources randomly but does not discard any resources.

### E.2 RESULTS

Table 5 shows that the results achieved by the altruistic agent trained with choice are equivalent to those achieved by the supervised agent. Furthermore, they are significantly better than those achieved by an agent with a random policy.

## F VIDEOS OF BEHAVIOUR OF ALTRUISTIC AGENT

We provide videos for the most relevant outcomes of our experiments in the supplementary material.

### F.1 VIDEOS FOR RESULTS OF GRIDWORLD EXPERIMENTS (SECTION 4.1)

#### F.1.1 DOOR SCENARIO IN FIG. 1 TOP CENTER

**01 Altruistic agent opens door for leader agent**: It can be observed that the altruistic agent has learned to operate the door switch to enable the leader agent to pass through the door and reach its target on the other side.

**02 Altruistic agent does not open door for leader agent (failure case)**: It can be observed that for an unfavourable choice of hyperparameters, the altruistic agent does not open the door.

#### F.1.2 DEAD END SCENARIO IN FIG. 1 TOP RIGHT

**03 Altruistic agent gives way to leader agent**: It can be observed that the altruistic agent does not get into the way of the leader agent, which is hence able to reach its target in the top right cell.

**04 Altruistic agent blocks path of leader agent (failure case)**: It can be observed that for an unfavourable choice of hyperparameters, the altruistic agent blocks the entry to the hallway towards the right side of the environment such that the leader agent cannot reach its target at the top right cell. This happens as the altruistic agent forcefully maximizes the estimated choice of the leader agent by hindering it from entering the hallway, which is a region of fewer estimated choice.

### F.2 VIDEO FOR RESULTS OF LEVEL BASED FORAGING (SECTION 4.2)

**05 Altruistic agent enables leader to forage apples**: It can be observed how the altruistic agent (blue) learned to coordinate its movements with the leader agent (green), to enable the leader agent to forage apples. It has learned this behaviour purely through optimizing for the leader agents choice and is itself not rewarded for foraging apples.

### F.3 VIDEO FOR RESULTS OF TAG (SECTION 4.3)

**06 Altruistic agents protect leader from adversaries**: It can be observed how the altruistic agents (blue colors) learned to coordinate their movements to protect the leader agent (green) from its adversaries. The adversaries (red colors) try to catch the leader, which in turn tries to flee from them. The altruistic agents protect the leader by actively intercepting the paths of the adversaries. They have learned this behaviour purely through optimizing for the leader agents choice.

