# OpenReview forum: "Learning Altruistic Behaviours in Reinforcement Learning without External Rewards"
_ICLR.cc/2022/Conference — ICLR 2022 Spotlight_

### Official Review · Reviewer_kHZT · 2021-11-02

**Correctness:** 4
**Technical Novelty And Significance:** 4
**Empirical Novelty And Significance:** 4
**Recommendation:** 8
**Confidence:** 3

**Main Review:**

I am not expert in the specified feild, but in my opnion:
1. The writing of this paper is generally well-organized and of good quality.
2. This paper, as far as I know, is the first to try to address this important problem.
3 This paper can serve as a baseline and also propose three testbeds for future research.

**Summary Of The Paper:**

This paper is the first try to train agent behave altruistically towards others without knowledge of their objective or any external supervision.
The main idea is to train altruist agent giving the leader agents more choice and thereby allowing them to better achieve their goals.  They  introduce three multi-agent environments of increasing complexity to evaluate the proposed method. The results show that it can, in some cases, outperform the supervised baselines.


**Summary Of The Review:**

A novelty work, worth accept.

---

> ### Author Response · Authors · 2021-11-22
> **Response**
>
> Thank you for your review. We are glad you found our work novel and significant. We are also happy you found the paper's writing clear.

---

### Official Review · Reviewer_U54a · 2021-11-02

**Correctness:** 3
**Technical Novelty And Significance:** 3
**Empirical Novelty And Significance:** 3
**Recommendation:** 6
**Confidence:** 4

**Main Review:**

=====Strengths=====

1. It is an interesting and novel way to train goal agnostic helping policy by maximizing the choice of other agents. The choice-based modeling is clearly motivated and defined, well situated in prior work, sufficiently compared against the similar yet different framework, empowerment.

2. The writing is very clear and the core ideas are conveyed successfully.

3. The experiments and discussions are thorough and informative. The results are promising.

4. The implementation was well documented and the details seem to be sufficient for reproducing the results.

=====Weaknesses=====

While the overall writing is clear and the experiments are comprehensive, I do however still have confusion/concerns about the comparison between this approach and the prior work, specifically i) empowerment and ii) MADDPG-like approach that learns other agents' policies. It seems to me that choice is similar to empowerment, with the addition of using agent-specific policy instead of arbitrary probing policy. If I understand this correctly, this means that empowerment only depends on the environment and is agent agnostic, while choice is agent dependent -- the definition of choice depends on a specific agent's policy. If this understanding is correct, then I have two questions:

Q1: Can you train a policy using a MADDPG-like method that jointly learns the policy of other agents and the policy of its own? Crucially, this is closer to your approach than the vanilla single agent DRL is, since it also explicitly depends on the policies of others.

Q2: Empowerment is agent agnostic, so in theory, it generalizes better to other situations where agents will have different policies than the ones seen during training. Could you comment on the generalization?


**Summary Of The Paper:**

This paper provides an alternative to empowerment for goal agnostic assistance RL policy training -- choice-based optimization. The core idea is to model possible subgoals as the choices of others and train policy to maximize the choices of other agents for achieving goal agnostic assistance. Evaluation against standard deep RL and the recent assistance via empowerment (AvE) method shows that the choice-based approach can be effective in multiple tasks.

**Summary Of The Review:**

I enjoyed reading this paper and I think the core idea is a valuable contribution to the multi-agent RL research. My main concern is about its comparison with empowerment and other related multi-agent RL work, which is detailed in my main review. I would appreciate some clarification from the authors and would be willing to increase my rating if my concern is addressed in the rebuttal and the revision.

---

> ### Author Response · Authors · 2021-11-19
> **Response 1**
>
> >Summary Of The Paper:
> This paper provides an alternative to empowerment for goal agnostic assistance RL policy training -- choice-based optimization. The core idea is to model possible subgoals as the choices of others and train policy to maximize the choices of other agents for achieving goal agnostic assistance. Evaluation against standard deep RL and the recent assistance via empowerment (AvE) method shows that the choice-based approach can be effective in multiple tasks.
> Main Review:
> =====Strengths=====
> It is an interesting and novel way to train goal agnostic helping policy by maximizing the choice of other agents. The choice-based modeling is clearly motivated and defined, well situated in prior work, sufficiently compared against the similar yet different framework, empowerment.
> The writing is very clear and the core ideas are conveyed successfully.
> The experiments and discussions are thorough and informative. The results are promising.
> The implementation was well documented and the details seem to be sufficient for reproducing the results.
> =====Weaknesses=====
> While the overall writing is clear and the experiments are comprehensive, I do however still have confusion/concerns about the comparison between this approach and the prior work, specifically i) empowerment and ii) MADDPG-like approach that learns other agents' policies. It seems to me that choice is similar to empowerment, with the addition of using agent-specific policy instead of arbitrary probing policy. If I understand this correctly, this means that empowerment only depends on the environment and is agent agnostic, while choice is agent dependent -- the definition of choice depends on a specific agent's policy. If this understanding is correct, then I have two questions:
>
> >Q1: Can you train a policy using a MADDPG-like method that jointly learns the policy of other agents and the policy of its own? Crucially, this is closer to your approach than the vanilla single agent DRL is, since it also explicitly depends on the policies of others.
>
> A1: Thank you, the relationship with MADDPG is worth elaborating on. Note that MADDPG is not a baseline for altruistic behaviour, as it does not itself state any objective for altruistic behaviour — it is a general MARL method that requires the agents' rewards to be defined externally, while our proposal is such a reward. In this sense, they are orthogonal proposals.
>
> Using a MADDPG-like approach to simultaneously learn the leader agent's and the altruistic agent's policy would be an interesting idea for future work. This could be done by using our proposed altruistic objective for one of the agents. However, note that it does not change the nature of our proposal.
>
> Please note that this joint training setup would though significantly change the experiment, as the leader agent would learn a policy that depends on the (continuously adapting) policy of the altruistic agent. Hence, the leader could learn a policy that ideally utilises the altruistic agent. Such a scenario would be similar to previous work in RL on so-called student teacher scenarios, where the leader (teacher) agent learns to optimally instruct the altruistic (student) agent (see for example [1] and [2]).
>
> [1] Hadfield-Menell, Dylan, et al. "Cooperative inverse reinforcement learning." Advances in neural information processing systems 29 (2016): 3909-3917.
> [2] Ho, Mark K., et al. "Showing versus doing: Teaching by demonstration." Advances in neural information processing systems 29 (2016): 3027-3035.

---

> > ### Comment · Reviewer_U54a · 2021-11-19
> > **Thanks for the response**
> >
> > Thanks for the response. When I say MADDPG-like baselines, I mean "fix the leader's policy and only train the helper's policy." Crucially, this means that the helper is jointly learning the policy of the leader agent as well as its own policy that depends on the leader agent's policy. So I see this as an alternative way of modeling and incorporating the leader's choices. It is less explicit than the method proposed in this paper, but it is not clear to me whether it would perform worse.

---

> > > ### Author Response · Authors · 2021-11-20
> > > **Reply**
> > >
> > > Thank you, this is indeed an interesting way to incorporate the leader's policy in our estimation of its choice. We justify our practical implementation in Section 4.2 ("Training" paragraph), which is motivated by our desire for simplicity. We believe that your suggestion can open up new variants in future work, including coordinating multiple altruistic agents, and so could easily stand on its own as another paper. We will strive to include such an experiment in the final version, but due to its complexity (including tuning) cannot do it within the discussion period.

---

> ### Author Response · Authors · 2021-11-19
> **Response 2**
>
> >Q2: Empowerment is agent agnostic, so in theory, it generalizes better to other situations where agents will have different policies than the ones seen during training. Could you comment on the generalization?
>
>
> A2: Empowerment is indeed an interesting objective for altruistic behaviour but has the following disadvantages:
>
> (1) It cannot be estimated from observation: In many RL scenarios, especially those related to real-world scenarios, only black-box access to the environment is given. To compute empowerment an environment simulator with white-box internal access is required, as rollouts of different policies of the leader agent would have to be evaluated at each time step but using the same environment state.
>
> (2): It is computationally demanding: To compute empowerment of the leader agent, Monte Carlo samples from many policies have to be evaluated. Even assuming access to an environment simulator, this often is computationally infeasible. For this reason, the authors of AvE also decided to work with an empowerment proxy.
>
> (3): It is still biased towards a manually-defined family of policies: The choice of family of policies to consider for the leader agent introduces a bias into the empowerment estimation. These difficulties were also pointed out by the authors of the AvE paper.
>
>
> >Summary Of The Review:
> I enjoyed reading this paper and I think the core idea is a valuable contribution to the multi-agent RL research. My main concern is about its comparison with empowerment and other related multi-agent RL work, which is detailed in my main review. I would appreciate some clarification from the authors and would be willing to increase my rating if my concern is addressed in the rebuttal and the revision.

---

> > ### Comment · Reviewer_U54a · 2021-11-19
> > **Thanks for the clarification**
> >
> > Thanks for the clarification. I agree with most of the disadvantages. However, I don't think (1) is necessarily true. Based on Algorithm 1 in the AvE paper, they could estimate the empowerment proxy in real world settings by doing random exploration, similar to many RL approaches that rely on an exploration phase. I don't see why it would have to require white-box access (unless I misunderstand something here?).

---

> > > ### Author Response · Authors · 2021-11-20
> > > **Response**
> > >
> > > In order to compute their objective at a time t, they must rerun the simulator multiple times from time t onwards, with different random policies. All of these rollouts must start with the exact same state at time t, but will produce different histories of future states, due to the random policies. All of this is used to compute the objective for a single time step t, and the same process must be repeated for all time steps. So, their method cannot be applied purely from observations of standard rollouts gathered with random policies; it requires resetting a simulator to particular times and states, which is white-box access that is not possible in the real world.

---

> ### Comment · Area_Chair_3gci · 2021-11-28
> **Update in score?**
>
> Hi,
>
> Thanks for participating in the discussion with the authors. Have you reached any conclusion? Do you want to maintain the same score or are you willing to update it?
>
> Area Chair

---

### Official Review · Reviewer_bgMP · 2021-11-08

**Correctness:** 3
**Technical Novelty And Significance:** 2
**Empirical Novelty And Significance:** 3
**Recommendation:** 8
**Confidence:** 3

**Main Review:**

The paper is technically sound but very hard to follow through. For instance, I found it very hard to find which baselines they had implemented until I referred to appendix in table 3.

I still don't understand what does "choice" of the agent mean and how it is different from sub-goals of the agent. I read it multiple times but its very confusing.

For the model free estimation of choice from observations, it is unclear how the altruistic agent estimates the leader agent’s policy entropy from observations. Can you provide the details such as the policy feature vector? FC layer? and a softmax layer? Additionally, what was the loss function used.

In the Unsupervised baseline (MaxEnt) implementation, it would be helpful to provide the entropy index since you are trying to maximize this entropy value over the state distribution of trajectory rollouts.


**Summary Of The Paper:**

The paper posits a method for agents that learns altruistic behaviour from observations and interactions with other agents without knowledge of their objective fucntion by estimating the sub-goals of the agent and actively maximizing their choice.

The authors define 3 different task agnostic method to estimate the choices made by the agents.

The experimental evaluation is extensive with testing the agents in 3 different environments while reporting the results in comparison with three baselines - Discrete Choice, Entropic Choice, and Assistance via Empowerment.


**Summary Of The Review:**

The authors are trying too many different(existing) methods to prove the central concept around how one agent can be altruistic to another by maximizing their reward. The paper is very hard to follow through. Although, the experiments are extensive.

---

> ### Author Response · Authors · 2021-11-19
> **Response**
>
> >Summary Of The Paper:
> The paper posits a method for agents that learns altruistic behaviour from observations and interactions with other agents without knowledge of their objective fucntion by estimating the sub-goals of the agent and actively maximizing their choice.
> The authors define 3 different task agnostic method to estimate the choices made by the agents.
> The experimental evaluation is extensive with testing the agents in 3 different environments while reporting the results in comparison with three baselines - Discrete Choice, Entropic Choice, and Assistance via Empowerment.
> Main Review:
> The paper is technically sound but very hard to follow through. For instance, I found it very hard to find which baselines they had implemented until I referred to appendix in table 3.
>
> A1: Thank you for this feedback. We now stated the used baselines more clearly in section 4 when describing the experimental details.
>
> >I still don't understand what does "choice" of the agent mean and how it is different from sub-goals of the agent. I read it multiple times but its very confusing.
>
> A2: Subgoals are not uniquely defined but can be seen as states that are desirable for an agent towards achieving its final goals. A state in which the agent has high choice can be seen as a state from which the agent can reach many possible other states. States of high choice can hence be understood as subgoals of intelligent agents (see Turner et al. (2019)); maintaining a high choice level is further correlated to other subgoals such as self-preservation or resource acquisition.
>
> >For the model free estimation of choice from observations, it is unclear how the altruistic agent estimates the leader agent’s policy entropy from observations. Can you provide the details such as the policy feature vector? FC layer? and a softmax layer? Additionally, what was the loss function used.
>
> A3: We used the original implementation from Christianos et al. (2020) and Lowe et al. (2017) (see section 3.2 and appendix A.2). These implementations use three fully connected layers of size 64 and an MSE loss.
>
> >In the Unsupervised baseline (MaxEnt) implementation, it would be helpful to provide the entropy index since you are trying to maximize this entropy value over the state distribution of trajectory rollouts.
>
> A4: Thank you for this input. We are not entirely sure what is referred to as the "entropy index", but we assume it is the state entropy. If so, we can visualize how the state entropy changes over the training duration (i.e. an additional line for state entropy in Figure 4). We are currently running this analysis and will update the Figure as soon as possible.
>
> >Summary Of The Review:
> The authors are trying too many different(existing) methods to prove the central concept around how one agent can be altruistic to another by maximizing their reward. The paper is very hard to follow through. Although, the experiments are extensive.
>
> A5: We appreciate that the reviewer found our experiments extensive and will happily clarify or improve any specific phrasing or equation that is felt to be unclear. We will also consider any argument for why a specific existing method should be left out.

---

> > ### Comment · Reviewer_bgMP · 2021-11-20
> > **Thanks.**
> >
> > The authors have clarified most of my main concerns. Thanks.

---

### Official Review · Reviewer_K6xu · 2021-11-08

**Correctness:** 3
**Technical Novelty And Significance:** 3
**Empirical Novelty And Significance:** 3
**Recommendation:** 6
**Confidence:** 4

**Main Review:**

## Strengths

1) I think the paper aims to tackle an important problem, namely that of inducing more altruistic behavior in artificial agents acting in the same environment with others and assisting others in achieving their goals without knowing what those are.

2) I also like the proposed approach because it seems quite general and doesn't assume access to the leader's reward function, state goal, policy, or trajectories (unlike other methods in this space).

3) The method is new as far as I know, although some of its elements have been used in other contexts, but I think the authors do a good job at explaining the connections to related work.

4) I also found the analysis of the different choice formulations from Figure 1 to be quite insightful.


## Weaknesses

### Baselines
1) In the paper, you write that for the AvE baseline, you used the hyperparameters suggested by the authors, but this doesn't seem fair since typically these methods need to be fine-tuned for each task / domain as it may require very different hyperparameters than the ones used in the original paper. Could you do a hyperparameter search for AvE and present the results with the best HPs found on the tasks used for evaluation?

2) It wasn't very clear to me why you are not using the AvE baseline for the LBF and Tag domains and also not using the Supervised baseline for the Gridworld tasks. Could you please add these for completion or explain in more detail why they are not used for comparison?

3) I think it would be useful to compare the methods with an oracle which would be the optimal policy for assisting the leader agent. This could provide insight into how far the current method is from such a policy and whether there is still potential for improvement on these tasks by future work.


### Clarity

1) One of my biggest concerns is that it seems like there might be some significant modes and implicit assumptions made by the method which are not openly discussed in the paper. First of all, it seems like the method assumes that the leader can solve the task during its training stage, while the altruistic agent is taking random actions. This implies that the altruistic agent doesn't need to learn a very complex behavior and is not necessary for the leader's success (even if it might help the leader achieve the goal sooner). A more realistic setting would be one in which the two are trained at the same time or at least, there are multiple training stages for both of them (that alternate).

Aren't there cases where the altruistic agent can hurt the leader's performance? For example, there might be dead end states in the environment which can be activate by the altruistic agent's actions but which would be better avoided by the leader. Given that the two agents are not training at the same time, the leader's policy may not be robust to such changes in the environment / new states, so they may end up in them if the altruistic agent aims to increase the number of reachable states. Would it be possible to bias the set of states enabled by the altruistic agent towards the set of states which are desirable for the leader to reach?

Could the authors confirm whether this understanding is correct and explain why they decided to not tackle this setting / train in this way? I think these issues should at least be discussed in more depth in the paper. It would also be great if the authors can train the method on a similar scenario (where the approach isn't necessarily expected to do well) to better understand the limitations of this approach and when it can be expected to be effective.

2) Another thing which is not openly discussed in the paper is the fact that it seems like your approach may actually require more training since you have two separate training stages (i.e. first you train the leader on the task and then you train the altruistic agent to assist the leader). Can you comment on the trade-off between final performance and learning efficiency and make this fact more transparent in the paper? It would be great to include a graph with performance as a function of the number of samples used for training for the entire training process, with a breakdown for the leader and altruistic agent.

3) At the beginning of the paper, it is not very clear what are the metrics you are looking to improve, so I suggest mentioning that in the introduction. Initially, it is not clear whether the altruistic agent is supposed to help with a) percentage of times the leader can solve the goal, b) number of steps needed to solve the goal, c) leader's sample / computational efficiency, or something else.


### Limitations

1) Could you extend this algorithm to non-deterministic environments? Would you just need to replace the unsupervised learning objective with empowerement?

2) Along similar lines, is it possible to extend the algorithm so that the two agents learn at the same time? This would involve dealing with the non-stationarity of the leader's policy. This seems like it would be a more general setting and might be in a better position to handle more challenging tasks, so it would be great to at least discuss it in the conclusion section.

**Summary Of The Paper:**

This paper proposes an unsupervised learning method for training an agent to assist another agent (called the leader) in solve its task (thus displaying a type of altruistic behavior) without access to the other agent's reward function or policy. The authors propose to use the notion of maximizing the leader's choice which is formalized as maximizing the number of different states the leader can reach at any point (within a given number of steps from its current state). The authors propose three variants of this method and evaluate them on three different domains, while also comparing them with other approaches.

**Summary Of The Review:**

I think the paper tackles an important and rather neglected problem, proposes a novel and quite general method for this setting, and demonstrate significant empirical gains on multiple tasks.

My main concern is that some of the potential failure modes of this approach are not discussed in great detail. Because the method is so general, there are many settings where it may not be the most effective approach (which, from my perspective, is not a problem in regards to publishing the paper as long as this point is clear to the readers). I would really like to see an experiment (or more) that aims to tackle a case where this method isn't expected to shine to 1) either openly illustrate the method's failure modes and where it shouldn't be used or perhaps 2) show that it still works decently well even in such settings. Another big concern is the comparisons with other baselines, including the tuning of AvE, and the lack of comparisons with AvE and the Supervised baseline for some of the tasks.

In conclusion, I am leaning towards rejecting this paper, but I am willing to increase my score if the authors can successfully answer the questions above and address my concerns (especially the ones mentioned in the above paragraph).

---

> ### Author Response · Authors · 2021-11-19
> **Response 1**
>
> >Summary Of The Paper:
> This paper proposes an unsupervised learning method for training an agent to assist another agent (called the leader) in solve its task (thus displaying a type of altruistic behavior) without access to the other agent's reward function or policy. The authors propose to use the notion of maximizing the leader's choice which is formalized as maximizing the number of different states the leader can reach at any point (within a given number of steps from its current state). The authors propose three variants of this method and evaluate them on three  different domains, while also comparing them with other approaches.
>
> >Main Review:
> Strengths
> I think the paper aims to tackle an important problem, namely that of inducing more altruistic behavior in artificial agents acting in the same environment with others and assisting others in achieving their goals without knowing what those are.
> I also like the proposed approach because it seems quite general and doesn't assume access to the leader's reward function, state goal, policy, or trajectories (unlike other methods in this space).
> The method is new as far as I know, although some of its elements have been used in other contexts, but I think the authors do a good job at explaining the connections to related work.
> I also found the analysis of the different choice formulations from Figure 1 to be quite insightful.
> Weaknesses
> Baselines
>
> >In the paper, you write that for the AvE baseline, you used the hyperparameters suggested by the authors, but this doesn't seem fair since typically these methods need to be fine-tuned for each task / domain as it may require very different hyperparameters than the ones used in the original paper. Could you do a hyperparameter search for AvE and present the results with the best HPs found on the tasks used for evaluation?
>
> A1: Thank you, we do agree that it is important to tune the methods properly to new settings. The only hyper parameter in the author's implementation is the (integer) horizon n. We added experiments that show the results for ranging from n=1 to n=12 in the appendix B.1.1., and we found that the results are stable across a large range.
>
> >It wasn't very clear to me why you are not using the AvE baseline for the LBF and Tag domains and also not using the Supervised baseline for the Gridworld tasks. Could you please add these for completion or explain in more detail why they are not used for comparison?
>
> A2: AvE requires privileged access to the environment, as a white-box simulation that supports restarts at arbitrary states. This is different from the other methods we compare, which simply act in a black-box environment, as in standard RL. Therefore it does not readily apply in the standard RL environments (LBF and Tag), and it would not be a fair comparison.
> However, we did try AvE in the LBF and Tag environments, but found that the computational demand of the many trajectory restarts yielded infeasible computation times for these larger-scale environments. As a reference, the environments tested in the AvE paper only have a single embodied agent, so are easier to learn.
> We also added the supervised baseline for the Gridworld environment, thank you for the suggestion (see Table 3 in the appendix).
>
> >I think it would be useful to compare the methods with an oracle which would be the optimal policy for assisting the leader agent. This could provide insight into how far the current method is from such a policy and whether there is still potential for improvement on these tasks by future work.
>
> A3: Although this would be insightful, generally an optimal oracle strategy cannot always be defined. For example in Tag, even with perfect knowledge of the rewards and environment characteristics, there is no trivial answer.
> An easier way to define a near-optimal altruistic agent is one that explicitly maximizes the reward of the leader agent (a setting similar to supervised cooperative RL). Hence, we believe that the "supervised" baseline serves this purpose well, and we now report it for all experiments.

---

> > ### Comment · Reviewer_K6xu · 2021-11-21
> > **Thank you**
> >
> > Thank you for the comprehensive answer. In particular, the clarifications on the setting you consider and the joint training experiments were very useful.
> >
> > However, I suggest further improving the clarity of the paper (which has been mentioned as a weakness by multiple reviewers) by including the answers provided during rebuttal. In particular, the problem setting and its assumptions need to be clarified and motivated.
> >
> > As most of my concerns have been adequately addressed, I am currently leaning towards recommending acceptance (assuming the authors will take this feedback into account).

---

> > > ### Author Response · Authors · 2021-11-23
> > > **Response**
> > >
> > > Thank you for your feedback, we are happy to hear that our clarifications were helpful. We will include all addressed topics in the final version of this work, as, unfortunately, only then the page limit will be increased to 10 pages.

---

> ### Author Response · Authors · 2021-11-19
> **Response 2**
>
> >Clarity
> One of my biggest concerns is that it seems like there might be some significant modes and implicit assumptions made by the method which are not openly discussed in the paper. First of all, it seems like the method assumes that the leader can solve the task during its training stage, while the altruistic agent is taking random actions.
>
> A4: While this would be true for joint training, it is not the case in our experiments. In LBF and Tag, the The leader agent is first trained in a fully-cooperative setting (not with a random agent), to achieve high proficiency. It is this well-performing leader that the altruistic agent must learn to cooperate with, without supervision. In Gridworld, it is possible to achieve a high proficiency of the leader agent also when the altruistic agent takes random actions during training, as Q-Learning is used and the state space is relatively small.
> Our motivation is to design agents that can eventually learn to cooperate with humans, who we generally take to be rational and have goal-directed behaviour. If the "leader agent" was being trained from scratch simultaneously, it would not act rationally (until it converges) and thus would not be a good proxy for a human leader, which is what we believe to be the most important application.
>
> >This implies that the altruistic agent doesn't need to learn a very complex behavior and is not necessary for the leader's success (even if it might help the leader achieve the goal sooner).
>
> A5: It is necessary, as we do compare to a random-policy baseline in both LBF and Tag. Our results show that the leader's performance is significantly worse in that case. Thus, effective altruistic behaviour is essential for the leader's success in the tested environments.
>
> >A more realistic setting would be one in which the two are trained at the same time or at least, there are multiple training stages for both of them (that alternate).
>
> A6: Thank you for this great input. As requested, we investigated the effects of joint training as an additional experiment. We tested joint training in the Gridworld experiments and found that our approach yields similar results as with two-phase training (see appendix B.3).
> In this research work, we purposely focus on the case where the leader agent's policy is fixed during training, i.e. we focus on assisting an agent with a fixed rational policy.
>
> In addition to the motivation from human behaviour mentioned in Q4, this removes a confounding factor, as joint training could yield unstable fixed points of the optimization (which is common in MARL), in which case we would not be assessing our objective, but the quality of the optimisation (which is orthogonal). Removing confounding factors allows us to validate our hypothesis more robustly.
>
> In addition, joint training is a different experimental setup, as the leader agent would learn a policy that depends on the (continuously adapting) policy of the altruistic agent. Hence, the leader could learn a policy that ideally utilizes the altruistic agent. Such a scenario would be similar to previous work in RL on so-called student teacher scenarios, where the leader (teacher) agent learns to optimally instruct the altruistic (student) agent (see for example [1,2]).
>
> [1] Hadfield-Menell, Dylan, et al. "Cooperative inverse reinforcement learning." Advances in neural information processing systems 29 (2016): 3909-3917.
>  [2] Ho, Mark K., et al. "Showing versus doing: Teaching by demonstration." Advances in neural information processing systems 29 (2016): 3027-3035.
>
> >Aren't there cases where the altruistic agent can hurt the leader's performance? For example, there might be dead end states in the environment which can be activate by the altruistic agent's actions but which would be better avoided by the leader. Given that the two agents are not training at the same time, the leader's policy may not be robust to such changes in the environment / new states, so they may end up in them if the altruistic agent aims to increase the number of reachable states.
>
> A7: Our understanding of this question is that the altruistic agent might enlarge the set of states that the leader agent can reach with potentially harmful dead-end states, which would better be avoided by the leader. If so, note that the altruistic agent does not have an influence on the leader agent's policy, hence such dead end states would only be pursued by the leader agent if its policy would prefer them over other available states. In summary, the assumption of a rational leader affords a degree of robustness against the altruistic agent's choice-expanding actions.

---

> ### Author Response · Authors · 2021-11-19
> **Response 3**
>
> >Would it be possible to bias the set of states enabled by the altruistic agent towards the set of states which are desirable for the leader to reach?
> Could the authors confirm whether this understanding is correct and explain why they decided to not tackle this setting / train in this way?
>
> A8: Generally, this would be a very good objective. It does require knowledge of which states are desirable to the leader, and thus its reward, while we focus on unsupervised settings where this is not known.
>
> >I think these issues should at least be discussed in more depth in the paper. It would also be great if the authors can train the method on a similar scenario (where the approach isn't necessarily expected to do well) to better understand the limitations of this approach and when it can be expected to be effective.
>
> A9: We would like to highlight our failure case analysis in Section 4.1., which showcases potential failure modes and how they can be addressed. In our conclusion, we do address this concern and we are convinced that the work constitutes a relevant novel approach to altruistic behaviour despite the limitations, and should facilitate future research in this area.
>
> >Another thing which is not openly discussed in the paper is the fact that it seems like your approach may actually require more training since you have two separate training stages (i.e. first you train the leader on the task and then you train the altruistic agent to assist the leader). Can you comment on the trade-off between final performance and learning efficiency and make this fact more transparent in the paper? It would be great to include a graph with performance as a function of the number of samples used for training for the entire training process, with a breakdown for the leader and altruistic agent.
>
> A10: As laid out in our answer to Q6, we purposely focus on assisting a leader agent with a rational policy. We therefore see the leader agent's policy as part of the experimental setup and do not consider the training process of the leader agent as part of our evaluations, which is in line with previous work (see e.g. the AvE paper). As mentioned in our answer to Q6, we added an analysis of joint training in the appendix, where we do not observe a relevant difference.
>
> As laid out in our answer to Q6, we purposely focus on assisting a leader agent with a rational policy. We therefore see the leader agent's policy as part of the experimental setup and do not consider the training process of the leader agent as part of our evaluations, which is in line with previous work (see e.g. the AvE paper). As mentioned in our answer to Q6, we added an analysis of joint training in the appendix, where we do not observe a relevant difference.
>
> >At the beginning of the paper, it is not very clear what are the metrics you are looking to improve, so I suggest mentioning that in the introduction. Initially, it is not clear whether the altruistic agent is supposed to help with a) percentage of times the leader can solve the goal, b) number of steps needed to solve the goal, c) leader's sample / computational efficiency, or something else.
>
> A11: Thank you for pointing this out. We adjusted our introduction in the reviewed submission. The objective of the altruistic agent is to maximize the reward of the leader without knowledge of its reward function, which we achieve by maximizing a proxy, the leader's choice. Thus it is equally applicable to a) or b), but we do not aim to improve the leader's learning, which would be c).
>
> >Limitations
> Could you extend this algorithm to non-deterministic environments? Would you just need to replace the unsupervised learning objective with empowerement?
>
> A12: Both the LBF and Tag environment are non-deterministic environments, and our approach delivers satisfying results in these. In general, both choice and empowerment can be used in deterministic and non-deterministic environments.
>
> >Along similar lines, is it possible to extend the algorithm so that the two agents learn at the same time? This would involve dealing with the non-stationarity of the leader's policy. This seems like it would be a more general setting and might be in a better position to handle more challenging tasks, so it would be great to at least discuss it in the conclusion section.
>
> A13: Thank you for this remark, we implemented this now; please refer to our answer to question Q6.

---

### Official Review · Reviewer_4g8o · 2021-11-08

**Correctness:** 3
**Technical Novelty And Significance:** 3
**Empirical Novelty And Significance:** 3
**Recommendation:** 6
**Confidence:** 3

**Main Review:**

Strong points of the paper:

The main idea is quite conceptually simple and an interesting approach to develop altruism for MARL.

The experimental section of the paper is well executed, and the analysis of the results is thorough. Particularly the choice estimate analysis for each environment is helpful for understanding why the method may help. The analysis of the failure case is also quite insightful.

Weak points of the paper:

The exposition in Section 3.2 about the theoretical support to the objective is rather nonspecific -- I feel that it would be helpful to introduce more specific, technical claims which follow from the arguments in Turner (2019)

While the results on both the discrete gate and the continuous environment are strong, they seem less surprising because they are both navigation-type environments where the value of a state should correlate well with the “choice” afforded by being at that state. The analysis in the results of the hide-and-seek environment mention that it may have outperformed the supervised baseline because it provided a denser reward than the sparse supervised “catching” signal. So, these are environments where it’s unsurprising that the choice heuristic would work well. Because the contribution of this paper is quite dependent on the empirical performance of the method, I think that it needs to be evaluated on additional settings where it is less clear that “choice” is directly correlated with the reward, to be convincing as a generally useful metric when the true environment reward is unavailable.

Questions:

In Figure 3, for the LBF environment, I can see how the leader agent should have low IC when waiting at the apple at the top and lower IC when it can choose either of the two apples at the bottom. However, to me this seems to contradict the point that IC is a good proxy for the leader agent receiving high rewards, because the altruistic agent needs to help the leader to harvest the apple regardless, and the altruistic agent is closer to the apple at the top?

What do the scores in Table 1 indicate? It is not described in the caption or in the text. Why is the score for LBF in percentages but it is not for Tag?


**Summary Of The Paper:**

This paper introduces a method for developing altruistic agents in a multi-agent RL (MARL) setting. The core idea is that an altruistic agent, in the absence of any further reward or goal information, may try to increase the “choices” for the agent it is cooperating with as a proxy. The paper argues that this is a suitable proxy for the unknown true reward function in many environments, because optimal policies tend to choose actions which lead to greater choice (larger coverage of state visitation) in the future, using analysis of instrumental convergence. The method is evaluated on discrete environments where the altruistic agent has to help open a gate, a level-based foraging environment, and a continuous state space tag environment, and the paper shows that the method can lead to altruistic behavior and improved rewards obtained by the leader agent in these settings.


**Summary Of The Review:**

The idea of this paper is conceptually simple and I quite like its application to this challenging setting of altruism without true rewards, as well as the analysis that the paper presents. However, mostly because of the choice of environments in the empirical evaluation, I feel that the paper hasn’t provided enough evidence for the claim of the applicability of this altruistic objective, and I think this is a borderline case.

---

> ### Author Response · Authors · 2021-11-19
> **Response 1**
>
> >Summary Of The Paper:
> This paper introduces a method for developing altruistic agents in a multi-agent RL (MARL) setting. The core idea is that an altruistic agent, in the absence of any further reward or goal information, may try to increase the “choices” for the agent it is cooperating with as a proxy. The paper argues that this is a suitable proxy for the unknown true reward function in many environments, because optimal policies tend to choose actions which lead to greater choice (larger coverage of state visitation) in the future, using analysis of instrumental convergence. The method is evaluated on discrete environments where the altruistic agent has to help open a gate, a level-based foraging environment, and a continuous state space tag environment, and the paper shows that the method can lead to altruistic behavior and improved rewards obtained by the leader agent in these settings.
>
> >Main Review:
> Strong points of the paper:
> The main idea is quite conceptually simple and an interesting approach to develop altruism for MARL.
> The experimental section of the paper is well executed, and the analysis of the results is thorough. Particularly the choice estimate analysis for each environment is helpful for understanding why the method may help. The analysis of the failure case is also quite insightful.
> Weak points of the paper:
> The exposition in Section 3.2 about the theoretical support to the objective is rather nonspecific -- I feel that it would be helpful to introduce more specific, technical claims which follow from the arguments in Turner (2019)
>
> A1: We understand that some confusion may have arisen because we describe Proposition 6.9 from v7 of their paper, which was the most recent at the time of our submission, but the new version v8 (to appear in NeurIPS 2021) reaches the same result without the concept of bottleneck states. We updated our paper to reflect this change (Section 3.2), and at the reviewer's request we now use the exact same technical terms that they use.
>
> Note that Turner et al.'s Proposition 6.9 is highly technical and not very self-contained -- it depends on several pages of definitions and lemmas, which we cannot hope to repeat formally even in a limited way. We believe that the original paper is the better source for the full technical detail, and we instead describe their result informally, which allowed us to compress it to half a page and relate it to our method.
>
> > While the results on both the discrete gate and the continuous environment are strong, they seem less surprising because they are both navigation-type environments where the value of a state should correlate well with the “choice” afforded by being at that state. The analysis in the results of the hide-and-seek environment mention that it may have outperformed the supervised baseline because it provided a denser reward than the sparse supervised “catching” signal. So, these are environments where it’s unsurprising that the choice heuristic would work well. Because the contribution of this paper is quite dependent on the empirical performance of the method, I think that it needs to be evaluated on additional settings where it is less clear that “choice” is directly correlated with the reward, to be convincing as a generally useful metric when the true environment reward is unavailable.
>
> A2: In many cases the choice is indeed correlated with various reward functions, which shows that it can be estimated without direct access to rewards; and this gives evidence of the generality of the method. In our failure case analysis in section 4.1 we demonstrate how even in simple grid environments we can find rewards that are not correlated with choice, with surprising and undesired behaviour of the altruistic agent. However, the results from Turner et al. (2019) provide theoretical reasoning that in many environments a majority of optimal agents do seek states of higher choice (see section 3.2), even if there will always be a minority where this is not the case.
>
> We also added a new experiment (see appendix E) where we test our approach in a resource management game, which is a non-spatial environment. We believe that this shows the generalization of our method to non-spatial environments, and would like to thank the reviewer for the suggestion.

---

> ### Author Response · Authors · 2021-11-19
> **Response 2**
>
> >Questions:
>
> >In Figure 3, for the LBF environment, I can see how the leader agent should have low IC when waiting at the apple at the top and lower IC >when it can choose either of the two apples at the bottom. However, to me this seems to contradict the point that IC is a good proxy for >the leader agent receiving high rewards, because the altruistic agent needs to help the leader to harvest the apple regardless, and the >altruistic agent is closer to the apple at the top?
>
> A3: This is a good question, as interpreting choice estimates is not always straightforward. Note that we are visualizing the leader's choice given that the altruistic agent is in its current position (on the far left). Since the altruist is far away, foraging the apple at the top is simply not an option, and so this position has low choice for the leader.
> Our choice estimate in Table 1 — which analyses the average choice over entire episodes — shows that in LBF choice is also higher if the agent is able to achieve higher rewards, so they are correlated.
>
> >What do the scores in Table 1 indicate? It is not described in the caption or in the text. Why is the score for LBF in percentages but it is >not for Tag?
>
> A4: Thank you for highlighting this, we changed this header in Table 1 to replace "score" by reward, i.e. the higher the better. Regarding your question on why the reward is in percentage only for the LBF environment: For LBF, the leader agent's reward can easily be normalised, as the maximum and minimum reward per episode are clearly defined (foraging all apples vs. foraging no apples). For Tag, the leader agent's reward cannot trivially be normalised as the leader agent receives a negative reward every time it is caught by the adversaries, hence the maximum and minimum are not clearly defined.
>
> >Summary Of The Review:
> The idea of this paper is conceptually simple and I quite like its application to this challenging setting of altruism without true rewards, as well as the analysis that the paper presents. However, mostly because of the choice of environments in the empirical evaluation, I feel that the paper hasn’t provided enough evidence for the claim of the applicability of this altruistic objective, and I think this is a borderline case.
>
> A5: We hope that our clarifications and the addition of a non-spatial environment alleviate this concern. We are aware of the limitations of our work, which we demonstrate in our failure case analysis and emphasise in our conclusion.

---

> > ### Comment · Reviewer_4g8o · 2021-11-27
> > **Response to authors**
> >
> > Thank you for the thorough response and clarifications. The addition of the non-spatial environment does help alleviate my concern, and I will adjust my score. As other reviewers brought up, integrating the clarifications into the paper will be quite helpful.

---

### Decision · Program_Chairs · 2022-01-20

**Decision:**

Accept (Spotlight)

**Comment:**

This work proposed a method for encouraging an agent showing altruistic behaviour towards another agent (leader) without having access to the leader's reward function. The basic idea is based on the hypothesis that having the ability to reach many future states (i.e., called choice) is useful for the leader agent, no matter what it reward function is. The altruistic agent learns a policy that maximizes the choice of the leader agent. The paper defines three notions of choice, and evaluates them on four environments.

The reviewers believe that this work attempts to solve an important problem, proposes a novel approach, and performs reasonably good experiments. The reviewers are all on the positive side at the end of the discussion phase. Therefore, I recommend acceptance of the paper. I also suggest a spotlight presentation for this work because of the novelty of the problem, which might be of interest to other researchers.

The authors have already done some revisions to their paper (including adding a new environment). I encourage them to consider any remaining comments from reviewers in their final version.